# Circular RNA hsa_circ_0051040 Promotes Hepatocellular Carcinoma Progression by Sponging miR-569 and Regulating ITGAV Expression

**DOI:** 10.3390/cells11223571

**Published:** 2022-11-11

**Authors:** Linling Ju, Min Yao, Rujian Lu, Yali Cao, Huixuan Wang, Liuxia Yuan, Feng Xiao, Jianguo Shao, Weihua Cai, Lin Chen, Zhaolian Bian

**Affiliations:** 1Medical School of Nantong University, Nantong University, Nantong 226001, China; 2Nantong Institute of Liver Disease, Affiliated Nantong Hospital 3 of Nantong University, Nantong Third People’s Hospital, Nantong University, Nantong 226006, China

**Keywords:** hsa_circ_0051040, hepatocellular carcinoma, miR-569, ITGAV, EMT

## Abstract

Accumulating evidence has demonstrated the roles of circular RNAs (circRNAs) in hepatocellular carcinoma (HCC); however, their roles in HCC need to be further studied. Through high-throughput human circRNA microarray analysis of HCC and adjacent normal tissues, we identified hsa_circ_0051040 as a novel candidate circRNA for the diagnosis and treatment of HCC. In this study, we found that hsa_circ_0051040 was overexpressed in HCC tissues and cell lines and that its expression was correlated with poor prognosis. Knockdown of hsa_circ_0051040 inhibited the migration, invasion, and proliferation of HCC cells in vitro and in vivo, whereas overexpression of hsa_circ_0051040 had the opposite effects. Moreover, our data demonstrated that hsa_circ_0051040 acted as a sponge for miR-569 to regulate ITGAV expression and induce EMT progression. Our findings indicated that hsa_circ_0051040 promotes HCC development and progression by sponging miR-569 to increase ITGAV expression. Thus, hsa_circ_0051040 is a good candidate as a therapeutic target.

## 1. Introduction

Hepatocellular carcinoma (HCC) is one of the most common primary liver malignancies and the second leading cause of cancer-related death worldwide [1]. Many patients with HCC have no obvious specific symptoms at the early stage, and HCC is discovered incidentally during liver disease follow-up or physical examination in combination with liver ultrasound and alpha-fetoprotein (AFP) measurement. When symptoms appear, HCC patients are most often diagnosed in the advanced stage. However, there are several treatment options for patients diagnosed at the advanced stage, but the frequencies of tumor metastasis and recurrence are high [2]. Therefore, it is urgent to explore novel biomarkers for the early diagnosis, prognosis, and treatment of HCC.

Circular RNAs (circRNAs), noncoding RNAs, are covalently linked to form a closed circular structure without a 5′ cap and 3′ poly(A) tail [3,4]. CircRNAs are more stable than their linear parental genes and are predominantly localized in the cytoplasm [3,5]. Therefore, circRNAs have great potential as valuable biomarkers for disease diagnosis and prognosis. Moreover, some studies have shown that circRNAs participate in various physiological and pathological processes, such as proliferation [6], apoptosis resistance [7], migration [8] and invasion [9]. Some circRNAs containing microRNA (miRNA) response elements can function as miRNA sponges to bind miRNAs specifically and alleviate the inhibitory effects of miRNAs on their downstream target genes [10]. Recently, increasing evidence has shown that aberrant expression of circRNAs is frequently observed in various cancers [11,12,13,14,15,16]. However, the function of circRNAs in the occurrence and development of HCC needs to be further investigated.

In our study, we analyzed the expression profile of circRNAs in HCC tissues through microarrays and identified a novel dysregulated circRNA, hsa_circ_0051040, whose expression was increased in HCC tissues. Additionally, the expression of hsa_circ_0051040 was closely related to HCC patient prognosis. We further demonstrated that hsa_circ_0051040 acted as a sponge of miR-569 to upregulate the expression of integrin alpha V (ITGAV), induce epithelial–mesenchymal transition (EMT) and consequently promote the proliferation, invasion and migration of HCC cells. Thus, hsa_circ_0051040 may be a novel therapeutic target for HCC.

## 2. Materials and Methods

### 2.1. Patient Tissue and Plasma Specimens

Fresh HCC tissues and adjacent nontumor sites were obtained from patients who were diagnosed with primary liver cancer prior to any therapy at Affiliated Nantong Hospital 3 of Nantong University between 2016 and 2020. The plasma and tissue samples were not obtained from the same set of individuals. Plasma samples of 186 HCC patients, 24 benign liver disease patients, and 122 healthy people were also obtained from Nantong Third People’s Hospital (Appendix A). The study was approved by the Ethics Committee of Affiliated Nantong Hospital 3 of Nantong University, and written informed consent was obtained from each participant. All samples were frozen immediately after collection and stored at −80 °C until further use.

### 2.2. Cell Culture and Transfection

The human HCC cell lines PLC/PRF/5, SK-HEP-1, HuH-7, Li-7, Hep3B2.1-7 and the normal human liver LO2 cell lines were purchased from the Chinese Academy of Sciences Cell Bank (Shanghai, China). PLC/PRF/5, SK-HEP-1 and HuH-7 cells were cultured in MEM (Gibco, Thermo Fisher Scientific, Inc., Waltham, MA, USA); Li-7 and Hep3B2.1-7 cells were cultured in RPMI 1640 medium (Gibco, Thermo Fisher Scientific, Inc., Waltham, MA, USA) supplemented with 10% fetal bovine serum (Gibco, Thermo Fisher Scientific, Inc., Waltham, MA, USA) in a humidified atmosphere of 5% CO_2_ at 37 °C.

### 2.3. CircRNA Microarray Analysis

CircRNA microarray analysis of five paired tissue samples (5 HCC tissues and 5 matched adjacent normal tissues) was performed with a CapitalBio Technology Human CircRNA Array v2. The CapitalBio Technology Human CircRNA Array v2 was designed with four identical arrays per slide (4 × 180 K format), with each array containing probes for approximately 170,340 human circRNAs. Each circRNA was simultaneously detected by a long probe and a short probe.

### 2.4. RNA Fluorescence In Situ Hybridization (FISH) and Nuclear and Cytoplasmic Extraction

Cy3-labeled probe sequences targeting hsa_circ_0051040 and FAM-labeled probes targeting miR-569 were constructed by GenePharma (Suzhou, China). Nuclei were stained with DAPI. The probe signals were detected by a Fluorescent in Situ Hybridization Kit (GenePharma, Suzhou, China) according to the manufacturer’s protocol. Images were acquired using fluorescence microscopy. Sequences of the FISH probes are listed in Appendix A. The nuclear and cytoplasmic fractions of RNA were extracted with an RNA Subcellular Isolation Kit (Active Motif, Carlsbad, CA, USA).

### 2.5. 5-Ethynyl-20-Deoxyuridine (EdU) Incorporation Assay

The EdU incorporation assay was carried out with an EdU detection Kit (RiboBio, Guangzhou, China) to assess cell proliferation viability according to the manufacturer’s protocol. All images were acquired with an Olympus IX73-FL-PH fluorescence microscope (Olympus, Tokyo, Japan). The experiments were performed at least three times independently with triplicate samples.

### 2.6. RNA Preparation and qRT-PCR

Total RNA was isolated using TRIzol reagent (Takara, Japan) according to the manufacturer’s protocol. The concentration of the purified total RNA was detected using a NanoDrop spectrophotometer (Thermo Fisher Scientific, Waltham, MA, USA). For mRNAs and circRNAs, cDNA was synthesized using a HiScript II 1st Strand cDNA Synthesis Kit (Vazyme, Nanjing, China). For RNase R treatment, 1 μg of total RNA was incubated with 0.1 μL RNase R (20 U/μL) and 1 μL 10× RNase R Reaction Buffer at 37 °C for 15 min. For miRNA, cDNA was synthesized using a miRNA reverse transcription PCR kit (RiboBio, Guangzhou, China). Genomic DNA (gDNA) was isolated with a QIAamp DNA Mini Kit (QIAGEN, Hilden, Germany). qRT-PCR was performed in a Bio–Rad CFX Real-Time PCR System. GAPDH was used as an endogenous control for circRNA and mRNA. U6 and 18S rRNA were used as endogenous controls for tissue and plasma miRNA, respectively. The relative RNA expression levels were calculated using the 2^−ΔΔCt^ method. Sequences of the qRT-PCR primers are listed in Appendix A. The experiments were performed at least three times independently with triplicate samples.

### 2.7. Western Blotting

Proteins were extracted in RIPA Lysis buffer (Beyotime, Shanghai, China) with protease and phosphatase inhibitors. Proteins were separated on 10% SDS-PAGE gels and then transferred onto nitrocellulose membranes (Millipore Corporation, Darmstadt, Germany). After blocking in nonfat milk, membranes were incubated overnight at 4 °C with a primary antibody and subsequently with a secondary antibody for 1 h at room temperature. The antibodies used in Western blotting were as follows: horseradish peroxidase (HRP)-conjugated anti-β-actin (1:5000 dilution, Proteintech, Wuhan, China), anti-ITGAV (1:2000 dilution, Abcam, Boston, MA, USA), anti-E-cadherin (1:2000 dilution, Proteintech, Wuhan, China), anti-Vimentin (1:2000 dilution, Proteintech, Wuhan, China), anti-Snail (1:2000 dilution, Proteintech, Wuhan, China), and HRP-conjugated goat anti-rabbit IgG (1:5000 dilution, Proteintech, Wuhan, China). The bands were visualized by an enhanced chemiluminescence detection system (Tanon, Shanghai, China), analyzed by ImageJ software and normalized to the internal control β-actin.

### 2.8. Wound-Healing Assay

For the wound-healing assay, transfected cells at a density of 100% were seeded into 6-well plates with serum-free medium. Streaks were created across the monolayer using a sterile 10 μL pipette tip. Images of cell migration were acquired 0 and 24 h after wounding using a microscope (Olympus, Tokyo, Japan). The experiments were performed at least three times independently with triplicate samples.

### 2.9. Colony Formation Assay

Transfected cells were plated in 6 cm dishes and fixed in 1% paraformaldehyde after incubation for 7 days. After staining with 0.1% crystal violet, cell colonies were counted and analyzed under a light microscope.

### 2.10. Dual Luciferase Reporter Assay

SK-HEP-1 and PLC/PRF/5 cells were seeded into 24-well plates and transfected with luciferase reporter vector (pmirGLO-hsa_circ_0051040-WT or pmirGLO-hsa_circ_0051040-Mut, pmirGLO-ITGAV-WT or pmirGLO-ITGAV-Mut) and the miR-569 mimic or miR-NC using Lipofectamine 3000 reagent. After 48 h of incubation, firefly and Renilla luciferase activities were quantified with a dual luciferase reporter assay (Promega) according to the manufacturer’s protocol. Firefly luciferase activity was normalized to Renilla luciferase activity.

### 2.11. Construction of Plasmids, SiRNAs, Lentiviral Vectors, and miRNA Mimics and Inhibitors

To construct the hsa_circ_0051040 overexpression plasmids, human hsa_circ_0051040 cDNA was synthesized and cloned into the pcDNA3.1 vector GenePharma (Suzhou, China). Empty vector was used as the negative control. SiRNAs targeting hsa_circ_0051040 (si1: GACTCATCCGTCGATGGTCT; si2: CCGTCGATGGTCTAGTCTAT; si3: CATCCGTCGATGGTCTAGT) and the corresponding negative control siRNA were synthesized by GenePharma (Suzhou, China). Hsa_circ_0051040 knockdown lentiviral vectors pGLV3-H1-GFP-Puro (LV-circ0051040) and the corresponding negative controls (LV-NC) were designed and synthesized by GenePharma (Suzhou, China). Mimics and inhibitor of miR-569 were synthesized by RiboBio (Guangzhou, China).

### 2.12. Biotin-Coupled Probe Pulldown Assay

Biotin-labeled hsa_circ_0051040 and a negative control probe were designed and synthesized by GenePharma (Suzhou, China). Briefly, hsa_circ_0051040-high-expressing PLC/PRF/5 and SK-HEP-1 cells were fixed with 1% formaldehyde for 10 min, and the cells were then lysed. After centrifugation, 50 μL of the supernatant was retained as input, and the rest was incubated with streptavidin-coated magnetic beads (M-280, Invitrogen, Oslo, Norway) conjugated with biotin-labeled probe overnight at 4 °C. The pull-down product was washed with wash buffer, and total RNA was then extracted to detect the expression of hsa_circ_0051040 and miRNAs by qRT-PCR. The experiments were performed at least three times independently with triplicate samples.

### 2.13. Cell Migration and Invasion Assays

Transwell assays were used to evaluate the invasion and migration abilities of cells in vitro. Briefly, transfected SK-HEP-1 or PLC/PRF/5 cells were plated in 24-well plates at a density of 1 × 10^5^ cells. Cells were resuspended in 200 μL of serum-free MEM medium and seeded into the upper chambers (Millipore, Darmstadt, Germany) with or without a Matrigel-coated membrane (BD Biosciences, Franklin Lakes, NJ, USA). MEM (600 μL) containing 20% FBS was placed into the bottom chambers as the attractant. After incubation for 48 h, the cells that migrated or invaded across the membrane were fixed with 4% paraformaldehyde, stained with 0.1% crystal violet solution for 30 min, and visualized under a microscope (Olympus, Tokyo, Japan). Cells were counted in five randomly selected microscopic fields. The experiments were performed at least three times independently with triplicate samples. The experiments were performed at least three times independently with triplicate samples.

### 2.14. H&E and Immunohistochemical (IHC) Staining

Paraffin sections (5 μm) of tissue samples were subjected to H&E and IHC staining. For IHC staining, paraffin sections were incubated with primary antibodies against Ki67 or ITGAV (1:100) overnight at 4 °C. After washing in PBS, the sections were incubated with anti-mouse HRP-conjugated secondary antibody for 1 h at room temperature. The paraffin sections were stained with DAB and hematoxylin and covered with coverslips for microscopic observation.

### 2.15. Animal Experiments

PLC/PRF/5 cells (1 × 10^7^) with stable knockdown of hsa_circ_0051040 or control cells resuspended in 100 μL of PBS were subcutaneously injected into the flanks of 6-week-old male BALB/c nude mice (8 nude mice were randomly divided into 2 groups, 4 in each group). Tumor growth was monitored every week by measuring the tumor width and length with calipers. Mice were sacrificed, and the tumors were excised, fixed with 4% paraformaldehyde, and processed for H&E and IHC staining. The liver metastasis model was established by spleen injection with two transfection groups of PLC/PRF/5 cells. After 30 days, we sacrificed the mice and evaluated the liver metastasis ability. H&E staining was performed to analyze the formation of metastasis. All procedures were approved by the Animal Care Committee of Nantong University.

### 2.16. Statistical Analysis

GraphPad Prism 7.0 (GraphPad, Inc., La Jolla, CA, USA) and SPSS version 17.0 software were used to conduct statistical analyses. Student’s *t* test was used for comparisons between two groups. Survival curves were constructed with the Kaplan–Meier method. Differences among three or more groups were analyzed by one-way analysis of variance (ANOVA). *p* values <0.05 were considered significant.

## 3. Results

### 3.1. hsa_circ_0051040 Is Overexpressed in HCC Tissues and Correlates with Poor Prognosis

To investigate the role of circRNAs, we performed a high-throughput human circRNA microarray to identify the differentially expressed circRNAs in HCC. A total of 170,340 circRNAs were detected in five pairs of HCC and adjacent normal tissues by circRNA microarray analysis. Box plots showing the circRNA profiles indicated similar distributions of all included samples after normalization in both groups (Figure 1A). The variation in these detected circRNAs between each group was assessed by heatmaps (Figure 1B) and volcano plots (Figure 1C). KEGG pathway analysis revealed the signaling pathways involved in the progression of HCC (Figure 1D). We focused on the upregulated circRNAs and screened the candidates according to the following criteria: (1) fold change ≥ 2, *p* < 0.05; (2) signal value greater than 1000; and (3) upregulated in each pair. Among these significantly upregulated circRNAs, hsa_circ_0051040 was selected as a candidate for further experiments. We examined the expression level of hsa_circ_0051040 in 91 HCC and paired adjacent normal tissues using qRT-PCR with divergent primers. The results showed that the hsa_circ_0051040 expression level in HCC tissues was significantly higher than that in paired adjacent normal tissues (Figure 1E) in up to 70% of the paired samples (Figure 1F). In addition, HCC tissues were divided into high-expression and low-expression groups according to the median hsa_circ_0051040 expression level. Kaplan–Meier analysis showed that HCC patients with a high hsa_circ_0051040 expression level had much shorter survival times than patients with a low hsa_circ_0051040 expression level (Figure 1G). The relationships between the hsa_circ_0051040 expression level and clinicopathological parameters were evaluated. The results indicated that the increased expression of hsa_circ_0051040 was associated with tumor size and lymph node metastasis (Table 1). To further investigate the clinical significance of hsa_circ_0051040 in HCC, FISH was performed on tissue microarray containing 98 HCC and paired adjacent normal tissues. The FISH results indicated that hsa_circ_0051040 expression level was markedly upregulated in HCC tissues compared with adjacent normal tissues (Figure 1H). To investigate the role of hsa_circ_0051040 as a biomarker for HCC, we detected the expression of hsa_circ_0051040 in plasma by qRT-PCR, and the results showed that the expression of plasma hsa_circ_0051040 was higher in patients with HCC than in individuals with benign liver diseases and healthy controls (Figure 1I). Receiver operating characteristic (ROC) curve analysis was conducted to assess the diagnostic sensitivity and specificity of hsa_circ_0051040 for HCC (Figure 1J). The area under the curve was 0.801, showing that hsa_circ_0051040 has a potential diagnostic capability. In addition, we detected the expression of plasma hsa_circ_0051040 before and after surgery. The results indicated that the postoperative expression of hsa_circ_0051040 was lower than that in patients with primary HCC (Figure 1K). In addition, analysis of clinicopathological parameters suggested that the expression of hsa_circ_0051040 was related to lymph node metastasis. We found that the plasma hsa_circ_0051040 expression level was elevated in the metastasis group compared with the non-metastasis group (Figure 1L).

### 3.2. Identification and Characteristics of hsa_circ_0051040 in HCC Cells

We investigated the expression level of hsa_circ_0051040 was significantly upregulated in HCC cells, among which the expression level of hsa_circ_0051040 was relatively high in SK-HEP-1 and PLC/PRF/5 cells and relatively low in Hep3B2.1-7 cells compared with LO2 cells (Figure 2A). Therefore, SK-HEP-1, PLC/PRF/5 and Hep3B2.1-7 cells were selected for further study. To further determine the cell distribution of hsa_circ_0051040, we detected the subcellular localization of hsa_circ_0051040 in HCC cells via FISH and nuclear and cytoplasmic extraction assay. The results indicated that hsa_circ_0051040 was predominantly localized in the cytoplasm (Figure 2B–D). We performed qRT-PCR using divergent primers for hsa_circ_0051040 containing the backsplice junction and convergent primers for both cDNA and gDNA. cDNA and gDNA extracted from SK-HEP-1 and PLC/PRF/5 cells were used as templates. The results showed that hsa_circ_0051040 was amplified by divergent primers only from cDNA but not from gDNA (Figure 2E). We noted that hsa_circ_0051040 was derived from exons 6 and 9 of fibrillarin (FBL) and located on chromosome 19q13.2. This circular product contained the head-to-tail splice junction of hsa_circ_0051040, which was amplified by qRT-PCR with divergent primers and confirmed by Sanger sequencing (Figure 2F). Furthermore, we confirmed that hsa_circ_0051040 was much more resistant to RNase R digestion than linear FBL mRNA (Figure 2G,H). These data confirmed that hsa_circ_0051040 was a circular RNA and stably localized in the cytoplasm.

### 3.3. hsa_circ_0051040 Promotes the Migration, Invasion and Proliferation of HCC Cells In Vitro

To explore the functions of hsa_circ_0051040 in HCC cells, two siRNAs targeting the backsplice region of hsa_circ_0051040 were transfected into SK-HEP-1 and PLC/PRF/5 cells (Figure 3A). The qRT-PCR results indicated that hsa_circ_0051040 expression was profoundly downregulated in SK-HEP-1 (Figure 3B) and PLC/PRF/5 cells (Figure 3C). We used Transwell and wound-healing assays to evaluate the migration and invasion abilities and used colony formation and EdU incorporation assays to evaluate the proliferation ability. The Transwell and wound-healing assays revealed that knockdown of hsa_circ_0051040 significantly suppressed the migration and invasion of SK-HEP-1 and PLC/PRF/5 cells (Figure 3D–F). The colony formation assays demonstrated that the colony numbers of SK-HEP-1 and PLC/PRF/5 cells were significantly decreased by the downregulation of hsa_circ_0051040 (Figure 3G). The EdU incorporation assays showed that knockdown of hsa_circ_0051040 decreased the percentage of EdU-positive cells (Figure 3H).

To further determine whether hsa_circ_0051040 upregulation has a positive cancer-promoting effect, we constructed the hsa_circ_0051040 overexpression vector pcDNA3.1-hsa_circ_0051040 and transfected the overexpression vector or the control vector into Hep3B2.1-7 HCC cells. The transfection efficiency was validated by qRT-PCR. The result from qRT-PCR indicated that hsa_circ_0051040 was upregulated over about 20 folds in Hep3B2.1-7 stable transfection (Figure 3I). Functionally, the Transwell and wound-healing assays revealed that the migration and invasion abilities of Hep3B2.1-7 cells were increased in the hsa_circ_0051040 overexpression group compared with the control group (Figure 3J,K). The colony formation assays showed that overexpression of hsa_circ_0051040 markedly increased the number of Hep3B2.1-7 cell colonies (Figure 3L). Moreover, the EdU incorporation assays further revealed that hsa_circ_0051040 overexpression increased the percentage of EdU-positive cells (Figure 3M). These results suggested that hsa_circ_0051040 significantly affected both the migration and invasion abilities and the proliferation ability of HCC cells.

### 3.4. hsa_circ_0051040 Acts as a Sponge of MiR-569 in HCC Cells

The potential target miRNAs of hsa_circ_0051040 were predicted by a bioinformatics approach. We used two separate databases (CirInteractome and circBank) to predict the potential miRNAs that bind to hsa_circ_0051040 and found seven miRNAs (miR-520h, miR-513a-5p, miR-532-3p, miR-545-3p, miR-545-5p, miR-569 and miR-924) from the intersection of the two databases (Figure 4A). Next, we used an RNA pulldown assay to elucidate whether hsa_circ_0051040 can directly bind these candidate miRNAs. RNA pulldown assays were conducted with a biotin-coupled hsa_circ_0051040 probe. The results of agarose gel electrophoresis indicated that hsa_circ_0051040 could be specifically enriched by the hsa_circ_0051040 probe compared with the oligo probe (Figure 4B). We purified the hsa_circ_0051040-bound RNA complex and analyzed the seven candidate miRNAs by qRT-PCR. The results showed that miR-569 was the only miRNA that was pulled down by hsa_circ_0051040 in both SK-HEP-1 and PLC/PRF/5 cells (Figure 4C,D). Co-localization of hsa_circ_0051040 and miR-569 in cytoplasm was observed (Figure 4E). We utilized FISH and qRT-PCR to assess miR-569 expression level and found that the miR-569 expression level was significantly downregulated in HCC tissues compared with adjacent normal tissues (Figure 4F,G). We constructed hsa_circ_0051040 luciferase reporter plasmids with wild-type (circ0051040-WT) or mutant (circ0051040-Mut) miR-569 binding sites (Figure 4H). SK-HEP-1 and PLC/PRF/5 cells were co-transfected with the circ0051040-WT or circ0051040-Mut luciferase reporter plasmid and the miR-569 mimic or NC. The results showed that the activity of the circ0051040-WT luciferase reporter was obviously reduced by the miR-569 mimic, while the activity of the circ0051040-Mut luciferase reporter was not affected by the miR-569 mimic (Figure 4I,J). These data suggest that hsa_circ_0051040 acts as a sponge for miR-569 in HCC cells.

### 3.5. hsa_circ_0051040 Promotes ITGAV Expression by Interacting with MiR-569 in HCC Cells

To explore the target gene of miR-569 that could be regulated by hsa_circ_0051040, PLC/PRF/5 cells transfected with hsa_circ_0051040 siRNAs (si1 and si2) were used for gene expression profiling to analyze the differentially expressed mRNAs. We then chose the 11 most downregulated genes for further target gene selection. We found that ITGAV was the target gene of miR-569, which was most significantly downregulated by hsa_circ_0051040 knockdown in HCC (Figure 5A). To determine the role of ITGAV in HCC progression, we evaluated ITGAV expression levels in HCC samples in The Cancer Genome Atlas (TCGA, https://cancergenome.nih.gov/, accessed on 7 February 2022) database. The results revealed that the ITGAV expression level was significantly higher in HCC patients than in normal controls (Figure 5B). The patients with high ITGAV expression levels had a shorter survival than those with low ITGAV expression levels by Kaplan–Meier analysis (Figure 5C). Moreover, Pearson correlation analysis indicated a positive correlation between hsa_circ_0051040 and ITGAV expression (Figure 5D). Since 3′UTR of ITGAV mRNA contains three potential binding sites for miR-569, we constructed luciferase reporter plasmids containing the wild-type (WT) or mutant-type (MUT) ITGAV 3′UTR transcript. (Figure 5E). The dual luciferase reporter assay showed that the miR-569 mimic significantly suppressed the activity of the WT but not the MUT luciferase reporter in both SK-HEP-1 and PLC/PRF/5 cells (Figure 5F,G). To further evaluate the effect of miR-569 on ITGAV expression, we transfected SK-HEP-1 and PLC/PRF/5 cells with the miR-569 mimic or inhibitor (Figure 5H,I) and detected the expression of ITGAV by Western blotting. The results showed that the miR-569 mimic significantly decreased the ITGAV protein level and that this decrease was reversed by the miR-569 inhibitor (Figure 5J). MiR-569 expression was negatively correlated with ITGAV expression, as determined by Pearson correlation analysis (Figure 5K). Moreover, qRT-PCR analysis demonstrated that the upregulation of circ_0051040 enhanced the relative expression of ITGAV and the effects caused by overexpressing circ_0051040 could be reversed by miR-569 mimic (Figure 5L). EMT plays a critical role in tumor cell migration and metastasis in HCC. Based on the role of hsa_circ_0051040 in the migration and invasion of HCC cells, we hypothesized that hsa_circ_0051040 may mediate the EMT process in HCC cells. We found that knockdown of hsa_circ_0051040 promoted the epithelial biomarker E-cadherin but inhibited the mesenchymal biomarkers Vimentin, Snail and ITGAV expression. Conversely, the overexpression of hsa_circ_0051040 significantly reduced the expression level of E-cadherin and enhanced the levels of Vimentin, Snail and ITGAV (Figure 5M). These findings indicated that hsa_circ_0051040 promoted the EMT process of HCC.

To verify whether hsa_circ_0051040 exerts its promotive effect on HCC by sponging miR-569, we further performed rescue assays to examine the functional interaction between hsa_circ_0051040 and miR-569. The results showed that the miR-569 inhibitor significantly reversed the inhibition of cell migration, invasion and proliferation induced by hsa_circ_0051040 knockdown (Figure 6A–C). These results indicated that hsa_circ_0051040 directly sponges miR-569 and that hsa_circ_0051040 knockdown suppresses HCC cell invasion and metastasis. Subsequently, we further assessed the effects of hsa_circ_0051040 and miR-569 on the EMT pathway. Western blot analysis showed that the ITGAV, Vimentin and Snail protein levels were decreased after hsa_circ_0051040 was knocked down but that this effect was partially abolished by the miR-569 inhibitor (Figure 6D). Hsa_circ_0051040 knockdown also restored the miR-569 inhibitor-induced overexpression of E-cadherin. The data indicated that hsa_circ_0051040 could act as a miR-569 sponge to regulate ITGAV expression and activate EMT progression.

### 3.6. hsa_circ_0051040 Knockdown Inhibits the Growth and Metastasis of HCC Cells In Vivo

To investigate the effect of hsa_circ_0051040 on tumor growth in vivo, PLC/PRF/5 cells transduced with NC lentiviral vectors (LV-NC) or hsa_circ_0051040 knockdown lentiviral vectors (LV-circ0051040) were subcutaneously injected into nude mice. The results revealed that the tumor volume in the LV-circ0051040 group was significantly reduced (Figure 7A). qRT-PCR analysis demonstrated that hsa_circ_0051040 (Figure 7B) and ITGAV (Figure 7C) expression was decreased in the LV-circ0051040 group compared with the LV-NC groups. We extracted protein from the tumors. Consistent with the above results, Western blot analysis showed that the expression level of ITGAV was decreased under the influence of LV-circ0051040 (Figure 7D). H&E staining indicated that the tumor cells in the LV-circ0051040 groups were significantly reduced. IHC analysis of tumors showed that downregulation of hsa_circ_0051040 decreased the protein expression levels of the proliferation marker Ki-67 and ITGAV (Figure 7E). The liver metastasis model was applied to study the role of hsa_circ_0051040 in EMT of HCC (Figure 7F). H&E staining suggested that knockdown of hsa_circ_0051040 remarkably inhibited metastasis of HCC into liver (Figure 7G,H). The number of metastatic foci was decreased with knockdown of hsa_circ_0051040 in the liver metastasis model (Figure 7I). Therefore, we concluded that knockdown of hsa_circ_0051040 inhibits HCC tumor growth and metastasis in vivo.

Taken together, these findings provide evidence that hsa_circ_0051040 regulates ITGAV expression through miR-569, contributing to the proliferation and metastasis of HCC (Figure 7J).

## 4. Discussion

Recently, studies concerning the functions of circRNAs have attracted increasing attention. Many studies have revealed a close correlation between circRNA expression and multiple diseases and pathological processes, especially tumorigenesis [16,17,18]. However, the roles and mechanism of circRNAs remain largely unknown. In this study, our circRNA microarray and qRT-PCR results showed that hsa_circ_0051040 was stably upregulated in both HCC tissues and cell lines. We found that high hsa_circ_0051040 levels were an independent risk factor for overall survival in HCC patients. In vitro and in vivo experiments showed that downregulated expression of hsa_circ_0051040 significantly inhibited cell migration, invasion and proliferation, indicating that hsa_circ_0051040 plays an oncogenic role in HCC cell growth. CircRNAs contain multiple predicted putative miRNA-binding sites and may act as ceRNAs for miRNAs [19]. FISH assays indicated that hsa_circ_0051040 and miR-569 were mainly colocalized in the cytoplasm of HCC cells. Moreover, we confirmed through bioinformatics analysis, biotin-labeled probe pulldown assays, and dual luciferase reporter assays that hsa_circ_0051040 can directly bind to miR-569. The abnormal expression of miRNAs in many human cancers plays an important role in tumorigenesis and metastasis [20,21,22,23]. MiR-569 has been reported to play a tumor suppressor role in lung cancer [24] and osteosarcoma [25]. However, the role of miR-569 in HCC has not been elucidated. In our study, we found that miR-569 expression was downregulated in HCC tissues. Importantly, miR-569 was indicated to reduce the expression of ITGAV by binding to the 3′-UTR of ITGAV using a dual luciferase reporter gene assay and Western blot analysis. The expression of miR-569 was negatively correlated with the expression of ITGAV. Further study showed that knockdown of hsa_circ_0051040 significantly decreased the expression of ITGAV in vitro and in vivo. The expression of hsa_circ_0051040 was positively correlated with the expression of ITGAV. Moreover, a previous study indicated that ITGAV was closely associated with the development of HCC [26,27]. It has been reported that ITGAV is overexpressed in HCC tissues and promotes migration and invasion by activating the EMT process. In this study, we observed that knockdown of hsa_circ_0051040 suppressed the EMT process in HCC cells, suggesting that the hsa_circ_0051040/miR-569/ITGAV axis might exert roles through promotion of EMT progression. Finally, our study showed that knockdown of hsa_circ_0051040 inhibited tumor growth and metastasis in vivo. Furthermore, Western blot analysis and IHC staining results showed that downregulation of hsa_circ_0051040 inhibited the expression of ITGAV in vivo.

Taken together, our findings showed that hsa_circ_0051040 functions as an oncogene and plays a notable role in the progression of HCC. Our study further revealed that the hsa_circ_0051040/miR-569/ITGAV axis is a new ceRNA regulatory network and promotes EMT progression in HCC. However, the role of the hsa_circ_0051040/miR-569/ITGAV axis in other cancers needs to be further investigated. In summary, our data demonstrated that hsa_circ_0051040 can sponge miR-569 to increase the expression of ITGAV and thereby promote EMT in HCC. These findings suggest that hsa_circ_0051040 might be a novel diagnostic and therapeutic target for HCC.

## Figures and Tables

**Figure 1 cells-11-03571-f001:**
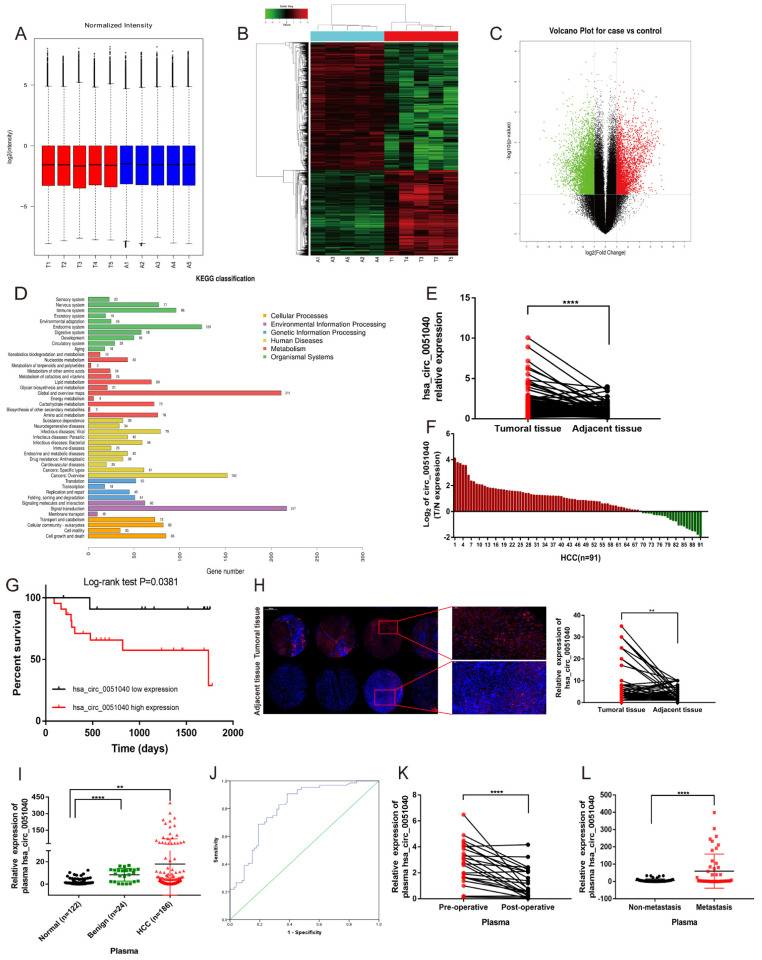
Upregulation of hsa_circ_0051040 in HCC. (**A**) Box plots showing that the distributions of circRNA profiles in both groups were nearly the same after normalization. HCC tissue samples: T1-T5; Adjacent normal tissue samples: A1–A5. (**B**) Heat map showing the upregulated and downregulated circRNAs in five pairs of HCC tissues (T) and adjacent normal tissues (A). (**C**) Volcano plot showing the upregulated and downregulated circRNAs in five pairs of HCC and adjacent normal tissues. Red points indicate upregulated circRNAs, green points indicate downregulated circRNAs, and black points indicate circRNAs with no significant difference. (**D**) KEGG analysis of circRNAs in five pairs of HCC and adjacent normal tissues. (**E**) qRT-PCR analysis of the fold change in hsa_circ_0051040 expression in paired HCC tissues compared with adjacent normal tissues. (**F**) The relative expression levels of hsa_circ_0051040 in HCC tissue samples (*n* = 91) were normalized to adjacent normal tissues. (**G**) Kaplan–Meier analysis indicated that patients with high hsa_circ_0051040 expression levels had much shorter survival. (**H**) FISH for hsa_circ_0051040 in 49 pairs of HCC and adjacent tissue microarrays. Cytoplasmic hsa_circ_0051040 was stained red, and the nuclei were stained blue with DAPI. (**I**) Detection of plasma hsa_circ_0051040 expression levels in patients with HCC (*n* = 186), patients with benign liver diseases (*n* = 24) and healthy controls (*n* = 122). (**J**) Construction of an ROC curve to compare the diagnostic performance of plasma hsa_circ_0051040 to discriminate HCC from healthy controls. (**K**) The expression levels of hsa_circ_0051040 were detected by qRT-PCR in 24 pairs of preoperative and postoperative patient plasma samples. (**L**) Plasma hsa_circ_0051040 expression levels were measured by qRT-PCR in HCC patients with metastasis and non-metastasis. Statistical significance was assessed by two-tailed Student’s *t* test. The error bar indicates the SD. ** *p* < 0.01; **** *p* < 0.0001.

**Figure 2 cells-11-03571-f002:**
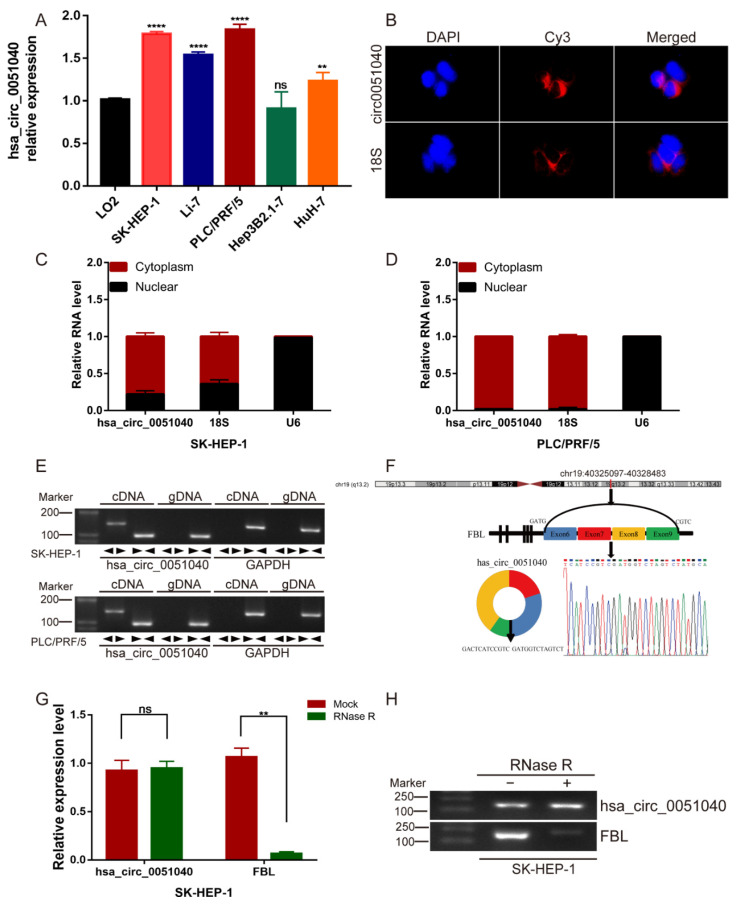
Identification and characteristics of hsa_circ_0051040 in HCC cells. (**A**) The expression levels of hsa_circ_0051040 were detected in HCC cell lines (SK-HEP-1, Li-7, PLC/PRF/5, Hep3B2.1-7 and HuH-7) compared to LO2 cells as control set as 1 by qRT-PCR. (**B**) FISH assay was used to detect the subcellular localization of hsa_circ_0051040 in HCC cells. hsa_circ_0051040 and 18S probe were labeled with Cy3. Nuclei were stained with DAPI. (**C**,**D**) qRT-PCR was used to measure the expression level of hsa_circ_0051040 in the nuclear and cytoplasmic of SK-HEP-1 and PLC/PRF/5 cells. (**E**) cDNA and gDNA of SK-HEP-1 and PLC/PRF/5 cells were used as templates to amplify hsa_circ_0051040 and GAPDH with divergent primers and convergent primers, respectively. As shown by agarose gel electrophoresis, hsa_circ_0051040 could only be amplified with divergent primers in cDNA but could not be amplified in gDNA. GAPDH was used as negative control. (**F**) Schematics illustrating that hsa_circ_0051040 is derived from FBL exons 6–9. The existence of hsa_circ_0051040 was proven by qRT-PCR, and its back splicing junction was verified by Sanger sequencing. (**G**) The expression of hsa_circ_0051040 and FBL mRNA levels were evaluated by qRT-PCR in SK-HEP-1 cells treated with or without RNase R. (**H**) Agarose gel electrophoresis of qRT-PCR amplification products. Statistical significance was assessed by two-tailed Student’s *t* test. The error bar indicates the SD. ** *p* < 0.01; **** *p* < 0.0001; ns, not significant.

**Figure 3 cells-11-03571-f003:**
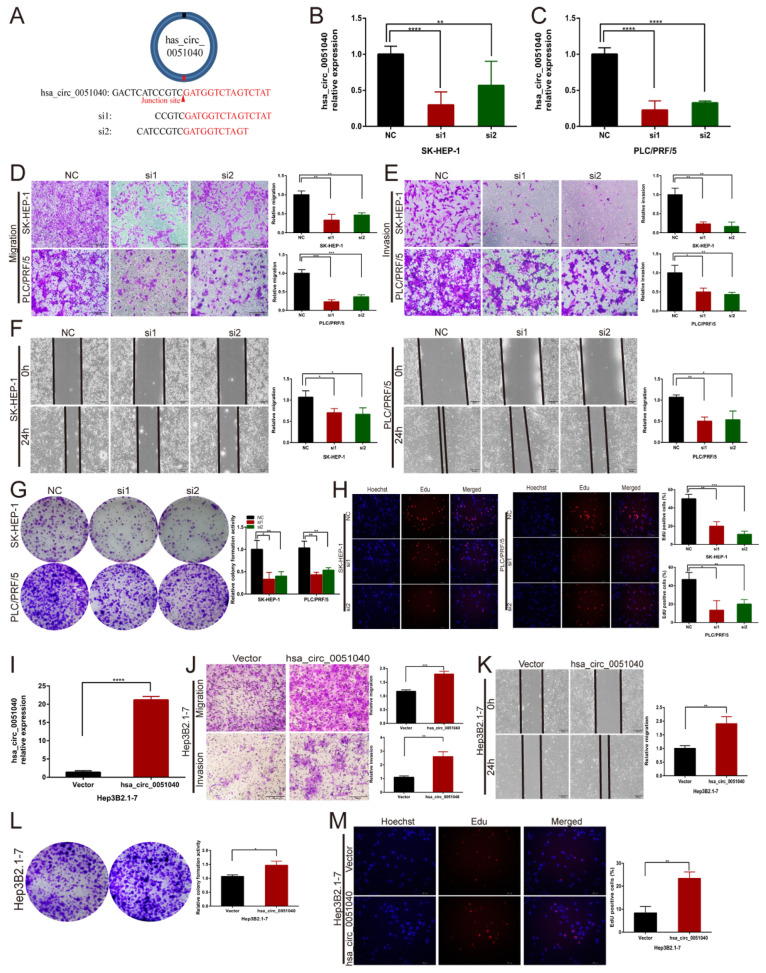
Hsa_circ_0051040 promotes the migration, invasion, proliferation and apoptosis of HCC cells in vitro. (**A**) Schematic illustration of two targeted siRNAs. siRNAs target the back splicing junction of hsa_circ_0051040. (**B**,**C**) qRT-PCR analysis of hsa_circ_0051040 mRNA in SK-HEP-1 and PLC/PRF/5 cells treated with siRNAs. (**D**,**E**) The migratory and invasive capabilities of hsa_circ_0051040 knockdown cells and corresponding controls were evaluated by Transwell assays (magnification, 20×; scale bar, 200 μm). (**F**) Wound-healing assays were performed to detect the migration of hsa_circ_0051040-knockdown transfected SK-HEP-1 and PLC/PRF/5 cells (magnification, 20×; scale bar, 200 μm). (**G**,**H**) The cell proliferation ability was assessed by colony formation and EdU incorporation assays after knocking down hsa_circ_0051040 in SK-HEP-1 and PLC/PRF/5 cells (magnification, 20×; scale bar, 200 μm). (**I**) qRT-PCR analysis of hsa_circ_0051040 mRNA in Hep3B2.1-7 cells treated with the hsa_circ_0051040 overexpression plasmid. (**J**) Cell migration and invasion abilities were determined by Transwell assays (magnification, 20×; scale bar, 200 μm). (**K**) Wound-healing assays were performed to detect the migration of hsa_circ_0051040 overexpression vector-transfected Hep3B2.1-7 cells (magnification, 20×; scale bar, 200 μm). (**L**,**M**) Cell proliferation detection of Hep3B2.1-7 cells was measured by colony formation and EdU incorporation assays. Statistical significance was assessed by two-tailed Student’s *t* test. The error bar indicates the SD. * *p* < 0.05; ** *p* < 0.01; *** *p* < 0.001; **** *p* < 0.0001.

**Figure 4 cells-11-03571-f004:**
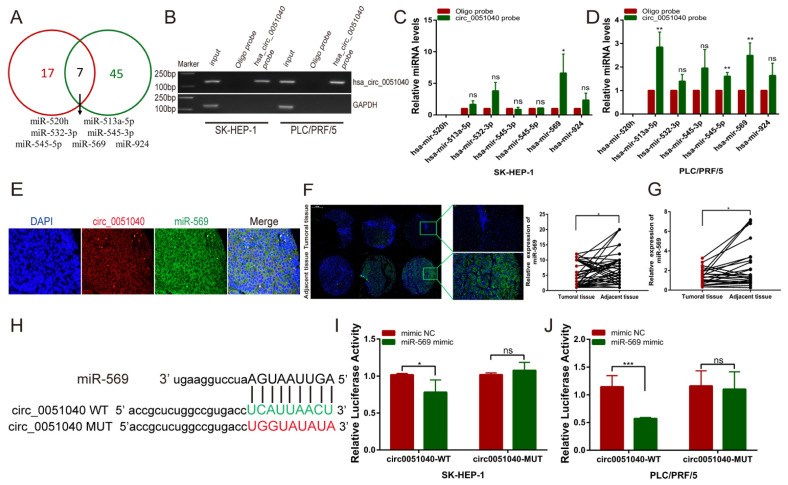
(**A**) Venn diagram analysis of the two databases’ (CirInteractome and circBank) identified seven miRNA candidates: miR-520h, miR-513a-5p, miR-532-3p, miR-545-3p, miR-545-5p, miR-569 and miR-924. (**B**) qRT-PCR and agarose gel electrophoresis results showed that hsa_circ_0051040 could be specifically pulled down by the hsa_circ_0051040 probe. (**C**,**D**) qRT-PCR analysis of the expression of 7 candidate miRNAs in the SK-HEP-1 and PLC/PRF/5 cell lysates after pulldown assay. Relative levels of hsa_circ_0051040 were normalized to input. (**E**) Co-localization of hsa_circ_0051040 (red color) and miR-569 (green color). The nuclei were stained with DAPI for blue color. (**F**) FISH for miR-569 in 49 pairs of HCC and adjacent tissue microarray. The cytoplasmic miR-569 was stained for green color, and the nuclei were stained with DAPI for blue color. (**G**) qRT-PCR analysis of miR-569 in paired HCC tissues compared with adjacent normal tissues. (**H**) Sequence alignment of binding sites between hsa_circ_0051040 and miR-569. (**I**,**J**) The relative luciferase activity of circ0051040-WT or circ0051040-MUT in the miR-569 mimic or NC group in SK-HEP-1 and PLC/PRF/5 cells. Statistical significance was assessed by two-tailed Student’s *t* test. The error bar indicates the SD. * *p* < 0.05; ** *p* < 0.01; *** *p* < 0.001; ns, not significant.

**Figure 5 cells-11-03571-f005:**
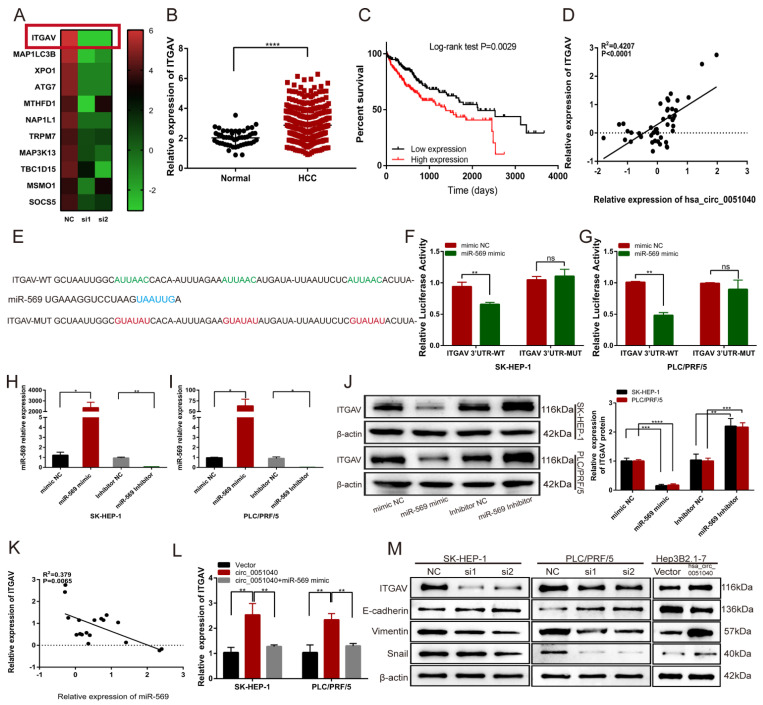
Hsa_circ_0051040 sponges miR-569 to upregulate ITGAV expression. (**A**) Heatmap analysis of the 11 most downregulated genes in this study. (**B**) ITGAV was upregulated in HCC according to TCGA. (**C**) Kaplan–Meier analysis indicated that patients with high ITGAV expression had low survival rates. (**D**) Correlation between hsa_circ_0051040 and ITGAV mRNA expression in HCC tissues. (**E**) Sequence alignment of binding sites between the ITGAV 3′UTR and miR-569. (**F**,**G**) The relative luciferase activity of ITGAV 3′UTR-WT or ITGAV 3′UTR-MUT in the miR-569 mimic or NC group in SK-HEP-1 and PLC/PRF/5 cells. (**H**,**I**) qRT-PCR analysis detected miR-569 expression levels in SK-HEP-1 and PLC/PRF/5 cells transfected with the miR-569 mimic or inhibitor. (**J**) ITGAV protein expression levels in SK-HEP-1 and PLC/PRF/5 cells transfected with the miR-569 mimic or inhibitor were detected by Western blotting. (**K**) Correlation between miR-569 and ITGAV mRNA expression levels in HCC tissues. (**L**) qRT-PCR analysis showed that the relative expression of ITGAV transfected with circ_0051040 plasmids and miR-569 mimic. (**M**) Protein levels of ITGAV, E-cadherin, Vimentin, and Snail in HCC cells with knockdown or overexpression of hsa_circ_0051040 were detected by Western blotting. Statistical significance was assessed by two-tailed Student’s *t* test. The error bar indicates the SD. * *p* <0.05; ** *p* <0.01; *** *p* < 0.001; **** *p* <0.0001; ns, not significant.

**Figure 6 cells-11-03571-f006:**
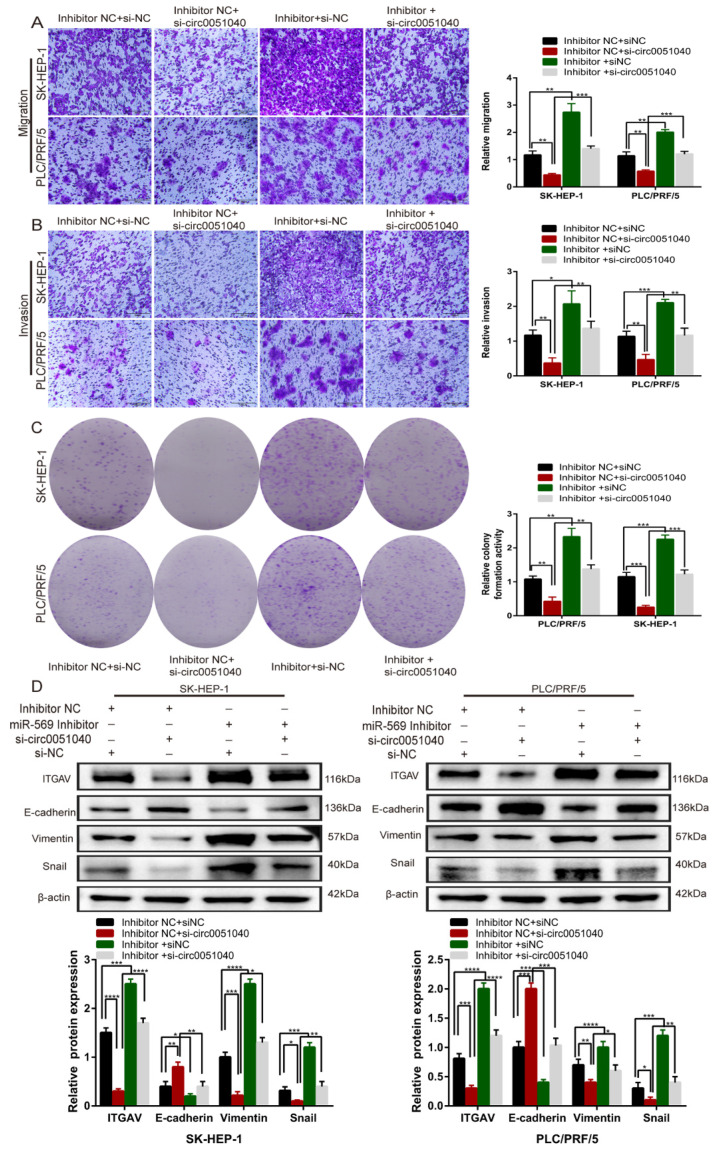
Hsa_circ_0051040/miR-569/ITGAV promoted EMT progression in HCC cells. (**A**–**C**) Representative images and quantification results of rescue migration, invasion and colony formation assays in SK-HEP-1 and PLC/PRF/5 cells transfected with circ0051040 siRNA and miR-569 inhibitor. Scale bar, 200 μm. (**D**) Western blot analysis of E-cadherin, Vimentin and Snail in SK-HEP-1 and PLC/PRF/5 cells after transfection with inhibitor NC + si-NC, inhibitor NC + si-circ0051040, inhibitor + si-NC and inhibitor + si-circ0051040. Statistical significance was assessed by two-tailed Student’s *t* test. The error bar indicates the SD. * *p* < 0.05; ** *p* < 0.01; *** *p* < 0.001; **** *p* <0.0001.

**Figure 7 cells-11-03571-f007:**
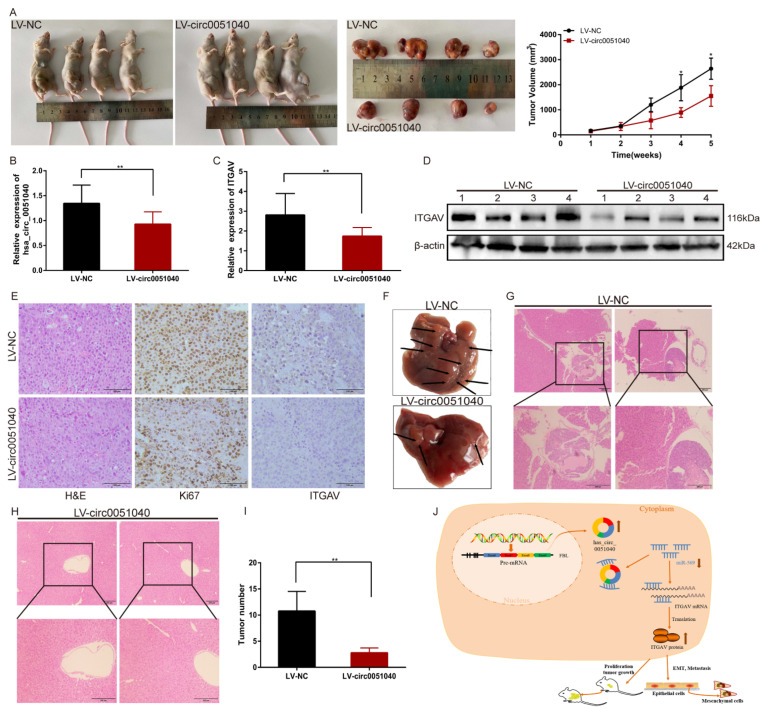
Knockdown of hsa_circ_0051040 suppresses tumor growth and metastasis in vivo. (**A**) Image of nude mice injected with PLC/PRF/5 cells subcutaneously (1 × 10^7^ cells per mouse, *n* = 4 mice per group). Tumors collected from mice were measured one month after subcutaneous injection. (**B**,**C**) The relative expression levels of hsa_circ_0051040 and ITGAV in tumors were determined by qRT-PCR analysis. The data are presented as the means ± SD. (**D**) Western blot analysis of ITGAV protein levels in dissected tumors transduced with hsa_circ_0051040 knockdown lentiviral vectors. (**E**) H&E staining showed the structure of tumors. Immunohistochemical staining of tumors for Ki-67 and ITGAV expression. Scale bar, 100 μm. (**F**) Representative images of tumor foci in the liver of two groups. (**G**,**H**) H&E staining images of the liver in two groups. (**I**) The number of tumor foci was calculated in two groups. (**J**) Schematic diagram of the regulatory mechanism of the hsa_circ_0051040/miR-569/ITGAV axis in promoting HCC proliferation and metastasis. The error bar indicates the SD. * *p* < 0.05; ** *p* < 0.01.

**Table 1 cells-11-03571-t001:** The clinicopathological parameters of HCC patients.

Characteristics	No.(%)	has_circ_0051040		*p* Value
		Low (%)	High (%)	
Gender				0.304
Male	66	29	37	
Female	25	14	11	
Age				0.875
<60	41	19	22	
≥60	50	24	26	
Tumor size (cm)				0.024
<5	59	33	26	
≥5	32	10	22	
TNM stage				0.178
T1-T2	16	10	6	
T3-T4	75	33	42	
Lymph node metastasis				0.002
Positive	20	3	17	
Negative	71	40	31	
Serum AFP (ng/mL)				0.156
<400	61	32	29	
≥400	30	11	19	

## Data Availability

Data are contained within the article.

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
