# Peer review of "Circular RNA hsa_circ_0051040 Promotes Hepatocellular Carcinoma Progression by Sponging miR-569 and Regulating ITGAV Expression"

_cells, 2022, doi:10.3390/cells11223571_

Round 1
Reviewer 1 Report
In this manuscript, Ju et al. studied the function of circRNA0051040 in hepatocellular carcinoma (HCC) progression. They performed microarray screening and identified circ0051040 is increased in HCC and correlated with poor prognosis. They further provided in vitro evidence showing circ0051040 promotes cell migration, invasion and proliferation. Mechanistically, the they proposed that circ0051040 sponges miR-589, further regulating ITGAV to promote HCC progression. My specific comments are as follows.
1. Multiple differential circRNAs identified by micro-array should be validated by qPCR to confirm the reliability of the micro-array assay.
2. What is the rationale to pick the has_circ_0051040 for deep study? Is the expression of circRNA dependent of its host gene?
3. Figure 2B, the signal looks more like staining signal instead of FISH signal. Plus, most of the circRNA signal is observed in nucleus in the upper panel, instead of in the cytoplasm as the authors stated in the manuscript.
4. Figure 3, the expression of the host gene should be examined to exclude the possible off-target effect of siRNAs.
5. The results in Figure 4H-J suggest circRNA is not sufficient to make the conclusion that the circRNA sponges miR-569. It is recommended to examine whether addition of WT circ0051040 attenuates the inhibition effects of miR569 on the UTR of its target genes, while addition of Mut circ0051040 not.
Reviewer 2 Report
The experiments of this study were designed to determine the role of hsa_circ_005104 in hepatocellular carcinoma progression and to identify downstream targets mediating its effects. The findings from this study make new insights into pathophysiology of hepatocellular carcinoma: 1) the study showed that hsa_circ_0051040 was overexpressed in HCC tissues and its expression was correlated with poor prognosis; 2) Knockdown of hsa_circ_0051040 inhibited the migration, invasion and proliferation of HCC cells; 3) the study demonstrated that hsa_circ_0051040 acted as a sponge for miR-569 to regulate ITGAV expression and induce EMT progression. Taken together, their findings reveal that hsa_circ_0051040 promotes HCC development and progression by sponging miR-569 to increase ITGAV expression. Overall, this is a very interesting study that has high novelty and potential significance in the field of non-coding RNA. The experiments were well-designed and the data are solid and clearly presented, and the findings from this study will be of interest to the journal’s readers. However, revision is required. The authors need to address a number of minor weaknesses as indicated below:
1. Line 237, the authors indicated that “hsa_circ_0051040 was predominantly localized in the cytoplasm”, however, in Figure 1H and Figure 2B, there was obvious localization of hsa_circ_0051040 in nucleus. High-quality images need to be provided to support the authors' conclusions.
2. The outliers in HCC group in Figure 1I are interesting, are the patients corresponding to these values different from the others in some way?
3. In Figure 2C, the products of convergent primer of hsa_circ_0051040 in SK-HEP-1 cells was different with PLC/PRF/5 cells, is this means that the host genes of the circRNA are different between these two cells?
4. In Figure 4E, the current images can not be clearly observed the co-localization of hsa_circ_0051040 and miR-569, especially in cytoplasm.
5. In Figure 5L, the authors used a circRNA plasmid. However, circRNA plasmid produced the circRNA in vivo containing circularized and non-circularized forms, is the circularized RNA the major form?
Round 2
Reviewer 2 Report
I appreciate the authors' revision and improvement of the manuscript.